# Improving the monitoring of deciduous broadleaf phenology using the Geostationary Operational Environmental Satellite (GOES) 16 and 17

Kathryn I. Wheeler[1], Michael C. Dietze[1]

[1]Department of Earth and Environment, Boston University, Boston, MA, 02215, USA

*Correspondence to*: Kathryn I. Wheeler (kiwheel@bu.edu)

**Abstract.** Monitoring leaf phenology tracks the progression of climate change and seasonal variations in a variety of organismal and ecosystem processes. Networks of finite-scale remote sensing, such as the PhenoCam Network, provide valuable information on phenological state at high temporal resolution, but have limited coverage. Satellite-based data with lower temporal resolution have primarily been used to more broadly measure phenology (*e.g.,* 16-day MODIS NDVI product).
Recent versions of the Geostationary Operational Environmental Satellites (GOES-16 and -17) can monitor NDVI at temporal scales comparable to that of PhenoCam throughout most of the western hemisphere. Here we begin to examine the current capacity of this new data to measure the phenology of deciduous broadleaf forests for the first two full calendar years of data (2018 and 2019) by fitting double-logistic Bayesian models and comparing the start, middle, and end of season transition dates to those obtained from PhenoCam and MODIS 16-day NDVI and EVI products. Compared to these MODIS products, GOES
was more correlated with PhenoCam at the start and middle of spring, but had a larger bias (3.35 ± 0.03 days later than PhenoCam) at the end of spring. Satellite-based autumn transition dates were mostly uncorrelated with those of PhenoCam. PhenoCam data produced significantly more certain (all *p*-values ≤ 0.013) estimates of all transition dates than any of the satellite sources did. GOES transition date uncertainties were significantly smaller than those of MODIS EVI for all transition dates (all *p*-values ≤ 0.026), but were only smaller (based on *p*-value < 0.05) than those from MODIS NDVI for the beginning
and middle of spring estimates. GOES will improve the monitoring of phenology at large spatial coverages and provides real-time indicators of phenological change even when the entire spring transition period occurs within the 16-day resolution of these MODIS products.

## 1 Introduction

The influence of leaf phenology is ubiquitous across many processes and relationships in ecology, local and regional climates,
and weather – ranging from leaf-trait relationships, nutrients in leaf litter leachate, surface roughness, transpiration, leaf-spectra relationships, albedo and energy budgets, and annual primary productivity (Alekseychik et al., 2017; Hudson et al., 2018; McKown et al., 2013; Piao et al., 2019; Richardson et al., 2012; Schwartz et al., 2002; Xue et al., 1996; Zhu and Zeng, 2017). In addition, since phenology is often highly sensitive to climatic variables such as temperature and precipitation (Killingbeck, 2004), it has been a primary ecological indicator of climate change (Parmesan and Yohe, 2003). Overall, spring in deciduous

forests has been found to advance and autumn has been found to delay (Gao et al., 2019; Liu et al., 2016), but the results are heterogeneous, especially for autumn (Gill et al., 2015; Richardson et al., 2013). These trends in changes are usually on the magnitude of days (*e.g.,* Keenan et al., 2014b, found an advancement in spring of $0.48 \pm 0.2$ days/year; Parmesan and Yohe, 2003, an advancement of 0.23 days/year). However, trends are often dependent on the phenology index used (Keenan et al., 2014b). This is particularly problematic for autumn where leaf color change often precedes leaf abscission, affecting the

similarity of autumn change in greenness indices (*e.g.,* Green Chromatic Coordinate; GCC), the Normalized Difference Vegetation Index (NDVI), and the Enhanced Vegetation Index (EVI). Similarly, observation frequency can be particularly important in spring, where the trend, the interannual variability, and time required for green-up can all be smaller than common satellite data product frequencies.

        The longest vegetation phenological records date back to the monitoring of the flowering of Japanese cherry trees in

the ninth century (Richardson et al., 2013). Since then many naturalists have tracked phenology in a variety of ecosystems, such as deciduous broadleaf (DB) forests. These human observations, though, are limited in scale and also rely on extensive manpower, time, and consistency. The United States National Phenology Network circumvents many of these challenges by relying on citizen scientist data, but is still limited by the timeliness of observation uploads and the inability to provide full, consistent coverage. Remote sensing techniques, both near-surface and satellite-based, monitor temporal changes in vegetation

reflectance at a near-real time and consistent frequency. Near-surface techniques include digital cameras, such as those that are part of the PhenoCam Network, that take repeated imagery of canopies and track how the ratios of red, green, and blue digital numbers change throughout the year (Richardson et al., 2007). The PhenoCam Network includes over 750 site-years of data across different biomes at the time of writing (Richardson et al., 2018b), but it has inherently limited spatial coverage.

        Satellites such as Aqua, Terra, Sentinel-2, and Landsat provide full coverage observations of phenological-sensitive

indices such as NDVI and EVI. However, in addition to being sensitive to clouds, these satellites are sun-synchronous (*i.e.,* their orbits are set to pass over a specific local time rather than being fixed over a specific location) and, thus, while observations have near global coverage, at any given site they are at a limited frequency. In addition, to cover the earth these orbits are not exactly the same each day and, thus, images are taken from varying viewing angles, which can add considerably complexity to the analysis and interpretation of data (*i.e.,* one needs to deconvolve changes in vegetation state from changes

in view angle). Because of this and challenges from frequent clouds, MODIS (Moderate Resolution Imaging Spectroradiometer, which is on Aqua and Terra) NDVI and EVI products are created by compositing data over multi-day periods (*e.g.,* 16 days). This can be interpolated into daily estimates of NDVI and EVI or can be provided as 16-day composites. While the daily interpolated products are sometimes used in phenology (*e.g.,* Ju et al., 2010; Keenan et al., 2014a; Liu et al., 2017), the 16-day composite NDVI and EVI products are also widely used (*e.g.,* Ahl et al., 2006; Hmimina et al., 2013;

Richardson et al., 2018b; Zheng and Zhu, 2017) and can be easily accessed though the MODIS web API and the MODISTools R Package (Tuck et al., 2014). This lower temporal resolution results in MODIS NDVI- and EVI- based estimates of phenological transition dates having larger uncertainties than those derived from PhenoCam (Klosterman et al., 2014). Additionally, while this temporal resolution may be adequate for some applications of NDVI, spring transitions and climate

change induced changes, as already mentioned, can happen at time scales much shorter than this 16-day resolution. This has
weakened how correlated the MODIS NDVI- and EVI- observed start of spring estimates are with those from PhenoCam (Filippa et al., 2018; Hufkens et al., 2012; Klosterman et al., 2014; Richardson et al., 2018a). To accurately track phenological transitions and changes at large scales, satellite-based data at a finer temporal resolution is needed.

The United States' National Oceanic and Atmospheric Administration's Geostationary Operational Environmental Satellite (GOES) -16 and -17 are the first satellites in the long-standing GOES series that possess a new sensor, the Advanced
Baseline Imager (ABI), that includes the necessary bands to calculate NDVI (Schmit et al., 2016). As the name implies, these satellites (one assumed the position of GOES-East at 75°W in December 2018 and one the position of GOES-West at 137°W in February 2019; Schmit et al., 2016) are geostationary and are thus not subject to many of the same limitations as sun-synchronous satellites because they take frequent measurements across their view with constant viewing angles. While geostationary satellites are still subject to clouds, the higher temporal resolution of potential measurements results in a greater
number of non-cloudy measurements than sun-synchronous satellites. GOES collects data every five minutes for the continental U.S. and every ten minutes for much of the western hemisphere. This high frequency data can be noisy in deciduous forests, however, but statistical models that utilize the characteristic diurnal pattern of NDVI can estimate daily midday NDVI values with uncertainty quantifications (Wheeler and Dietze, 2019). Additional geostationary satellites that possess the ability to monitor NDVI over other parts of the world include Meteosat over Africa and Europe, and Himawari over east Asia and
Oceania.

In this study, we investigated how GOES-16 (and by association GOES-17) compares to commonly-used 16-day MODIS NDVI and EVI products in relation to PhenoCam through estimations of phenology transition dates for DB forests in the eastern U.S. We selected sites within the PhenoCam Network and fit phenological curves for the different data sources (PhenoCam, MODIS NDVI, MODIS EVI, and GOES NDVI) in a Bayesian context for the first full calendar years of data
(2018 and 2019). We calculated start, middle, and end of season transition dates and compared those estimates between the different data sources. We hypothesized (1) GOES's higher measurement frequency would generate spring transition date estimates that are more similar than MODIS to PhenoCam; (2) since DB canopy spring transitions often occur faster than autumn, and changes in leaf color and area are more synchronous, spring transition dates would be more similar across the different data sources than the autumn ones; (3) since there exist differences in the sensitivities of different sensors to leaf
color versus leaf presence, GOES autumn transition dates would be most similar to MODIS NDVI; and (4) because of the higher data volumes, GOES would produce transition date estimates with lower uncertainties than MODIS.

## 2 Methods

### 2.1 Site selection

From the PhenoCam Network, fifteen DB sites were selected to be compared to their associated MODIS and GOES pixels. To
attempt to maintain homogeneity in the associated pixels of different spatial resolution (especially since the MODIS pixels do

not necessarily fall completely within the GOES pixels), we used Google Earth to exclude PhenoCam sites that were within the width of a GOES pixel (~1 km) from another land cover type (*e.g.,* grassland, urban, or large water body). Distinguishing evergreen species using Google Earth is more difficult and, thus, several of the sites do likely have nearby evergreen species that are included in the same satellite pixels. However, these sites still display predominantly DB phenology curves and, thus, were still included in this comparison. Specific site locations are given in Fig. 1 and Table 1. Additional metadata on the sites is available on the PhenoCam Network website (https://phenocam.sr.unh.edu/webcam/).

## 2.2 Data processing

### 2.2.1 GOES data download and quality control

The study time period of 1 January 2018 through 31 December 2019 was selected due to the availability of new GOES data for the first two calendar years.

ABI L1b radiance values ("CONUS" coverage region) were downloaded from NOAA's Comprehensive Large Array-data Stewardship System for the GOES channel 3 (near infrared) and channel 2 (red) for the study period (GOES-R Calibration Working Group and GOES-R Series Program, 2017). After the ABI L2+ clear sky mask (ACM) and data quality flags were applied (GOES-R Algorithm Working Group and GOES-R Series Program, 2018), radiance values were converted to reflectance factors under the guidance of the GOES R Product Definition and Users' Guide (https://www.goes-r.gov/users/docs/PUG-L1b-vol3.pdf;

 pp 27-28). Additional information on accessing and processing the data are given in (Wheeler and Dietze, 2019) and through our Github repository https://github.com/k-wheeler/NEFI_pheno/tree/master/GOESDiurnalNDVI. To account for differences in spatial resolution, the four 0.5 km red reflectance factors that fell within a near infrared (NIR) pixel were averaged together and NDVI was calculated on a 1 km spatial resolution calculated following Eq. (1):

$$NDVI = \frac{\rho NIR - \rho Red}{\rho NIR + \rho Red},\qquad\qquad(1)$$

where $\rho NIR$ and $\rho Red$ refer to the reflectance factor at the NIR band and red bands, respectively. While GOES does provide a blue band, which would allow for the calculation of EVI, additional calibration would likely need to be conducted to establish coefficients needed in the EVI equation, which is outside of the scope of this study. NDVI values that occurred before 1.5 hours after sunrise and after 1.5 hours before sunset (calculated using the Suncalc R package; Thieurmel and Elmarhraoui, 2019) were removed due to high noise. Additionally, the NDVI values of 0.6040 were regularly and abnormally present in the dataset early in the morning or in the evening throughout the study period and, thus, were removed as noise (*e.g.,* Supplementary Fig. S1). All calculations were performed in R (R Core Team, 2017).

## 2.2.2 Daily GOES NDVI estimates

Daily midday GOES NDVI values were estimated using the Bayesian statistical model described in (Wheeler and Dietze, 2019). In summary, this model relies on the characteristic diurnal NDVI pattern for DB pixels of increasing in the morning (represented with an inverted exponential decrease function) and decreasing in the afternoon (represented with an exponential decrease function), with a changepoint parameter between the two exponential functions (Supplementary Fig. S2). The error model accounts for negative bias in noise due to atmospheric attenuation (*e.g.,* from clouds and aerosols) by calculating the

probability that each observation is clear or cloudy and the amount of atmospheric transmissivity. Daily midday NDVI values with 95% credible intervals (CI) were obtained from the GOES data for all days with at least ten observations (*i.e.,* observations where both radiance values had data quality flags of "acceptable" and had an "acceptable" non-cloudy value from the ACM product). We changed the prior on the parameter $c$ (the midday maximum NDVI estimate) from that reported in Wheeler and Dietze (2019) to an uninformative Beta(1,1) instead of Beta(2,1.5). With more data, it was clear that the original prior was

incorrectly pulling fits for winter days too high. We also filtered out days that had < 25 observations and did not have observations in at least five different hours. These thresholds were determined by examining various combinations to minimize erroneous data and maintain most of the fitted days. Additionally, after a visual inspection of the diurnal fits and data, 68 days of the total 3645 days (< 2%) that remained after the previous filtering were removed due to poor fits that were heavily influenced by one outlier point. As this research scales up, implementing and testing automatic quality control methods will

become more important, but that is beyond the scope of this paper as here we want to focus on assessing phenological patterns and not remote sensing quality control algorithms.

Because overall atmospheric attenuation and differences in viewing angles between the sites were not corrected for, the NDVI values are not exactly equivalent to those measured at the ground. Thus, while we show comparisons across days at the same site is possible, we advise further NDVI processing is needed to make comparisons of NDVI values across sites or

use the underlying radiances in radiative transfer models. Transition dates and the seasonal curves for individual sites should be minimally affected because the view angle remains constant within a site and the processing algorithm does account for subdaily variability in atmospheric attenuation.

## 2.2.3 MODIS and PhenoCam data download

The 250m 16-day NDVI and EVI bands from the MODIS product MOD13Q1 were used for the study period (Didan, 2015).

This product was selected because it is easily accessible though the MODISTools R Package (Tuck et al., 2014) and the same temporal resolution for MODIS products has been used in numerous other comparisons between phenology data sources (*e.g.,* Ahl et al., 2006; Hmimina et al., 2013; Richardson et al., 2018b; Zheng and Zhu, 2017). Product data quality flags were applied to MODIS data. Daily midday PhenoCam GCC and standard deviation values were downloaded directly from the PhenoCam website archive (PhenoCam, 2018 and 2019; Seyednasrollah et al., 2019). The PhenoCam at Russell Sage did not collect data

for most of 2019 and, thus, we only fit one year of data (2018) to this site.

## 2.3 Phenology model fitting

Phenological curves were fit for each source of data (GOES NDVI, MODIS NDVI, MODIS EVI, and PhenoCam GCC). Based on the highly cited Zhang et al. (2003) paper, spring and autumn phenological changes were both modeled using a logistic curve calculated following Eq. (2):

$$\mu_t = \frac{c}{1 + e^{a+bt}} + d \,, \tag{2}$$

where $t$ is the time in days, $\mu_t$ is the phenology metric, $a$ and $b$ are fitting parameters, $c + d$ is the maximum value, and $d$ is the winter background value for the metric. The interpretation and realistic limits of the parameters $a$ and $b$ are somewhat obscure, so we reparametrized the model in terms of the midpoint date (50% change), $M = -a/b$. This gives a double-logistic curve calculated following Eq. (3):

$$\mu_t = \begin{cases} \frac{c}{1 + exp\,[b_A(t - M_A)]} + d & t > k \\ \frac{c}{1 + exp\,[b_S(t - M_S)]} + d & t \leq k \end{cases}, \tag{3}$$

where $b_A$ and $b_S$ indicate the $b$ parameters (rate of change) for the autumn green-down and the spring green-up, respectively; $M_A$ and $M_S$ are the autumn and spring midpoints, respectively; and $k$ is the change-point day in the summer where the function switches from the green-down logistic curve to the green-up logistic curve, which we assumed to be the 182[nd] day of the year (1 July) in order to separate the year into two, which has been done elsewhere (*e.g.,* Fu et al., 2016). Spring green-up was completed by the end of June in all of our sites; thus, the model fits were not heavily sensitive to this assumption. We assumed that the minimum ($d$) and the maximum ($c + d$) phenological index values (GCC, NDVI, EVI) would not change during this one year for all sites and, thus, both parts of the change-point function fit the same $c$ and $d$ values. Years were fit independently for each site and, thus, we did not assume that the $c$ and $d$ values were the same between years. To compare the influence of the different data sources on the uncertainty in the posterior, we used relatively uninformative Gaussian priors for the parameters (Table 2), which were created through simulating reasonable data. Since the motivation for this study was to illustrate the ability of GOES to monitor phenological change by comparing it to other remotely-sensed data sources, we focused on only one transition date estimation method; though, additional methods are explored elsewhere (*e.g.,* Klosterman et al., 2014)

Additionally, since the diurnal fit method (Wheeler and Dietze, 2019) produces estimates of uncertainty on daily NDVI, the means and precisions were incorporated in a normally-distributed errors-in-variables model within the phenology model. Likewise, errors-in-variables models were applied for the PhenoCam sites using the provided daily GCC standard deviation. Since the MOD13Q1 product does not provide a daily quantification of uncertainty (other than the quality flags that

we separately applied), generic values were used based on the standard deviations given in Miura et al., (2000) of 0.01 and 0.02 for NDVI and EVI observations, respectively.

The phenology models were fit in JAGS (Plummer, 2003; version 4.3.0) using standard Markov Chain Monte Carlo (MCMC) approaches. JAGS was called from R (R Core Team, 2017; version 3.4.1) using the rjags (Plummer, 2018; version 4.7) and runjags (Denwood, 2016) packages. Five chains were run and all models converged as assessed using Gelman-Brooks-Rubin statistic (GBR < 1.05) and all had effective sample sizes > 5000 after burn-in was removed. From the posterior outputs, 95% CI's were calculated. To plot and compare the different data sources, which inherently have different ranges, the predicted phenological curves for all joint parameter posteriors were rescaled to have a range of 0 to 1.

As explained in Zhang et al. (2003), the start and end of season transition dates for the double logistic fits was calculated as the roots of the third derivative of Eq. (3), which is illustrated in Fig. 2. The roots of the third derivative of our reparametrized function (Eq. 3) were calculated following Eqs. 4 and 5:

$$Root1 = \frac{b \times M + log\ (\sqrt{3}+2)}{b}, \tag{4}$$

$$Root2 = \frac{b \times M + log\ (2-\sqrt{3})}{b}, \tag{5}$$

where *Root1* signifies the start of season and end of season transition dates for spring and autumn, respectively. *Root2* signifies the end of season and start of season for spring and autumn, respectively. 95% CI's were calculated from the 50% midpoint transition date posteriors and from the sample-specific *Root1* and *Root2* values.

## 2.4 Model comparison

Transition dates were compared between the different data sources with an emphasis on how the transition date estimates from the satellite-based data compared to PhenoCam. We calculated the coefficient of determination ($R^2$) and root mean square error (RMSE) by comparing the means of the transition dates from one data source with the means of the transition dates from another source for all possible combinations. It is important to note that these calculations are based on the deviation from the one-to-one line and not the line-of-best fit, which is often different from the predicted line. Since we are testing the similarity of the transition date estimates from the two sources, the predicted line is the one-to-one line. Additionally, bias of the transition dates was assessed by subtracting samples from the joint parameter posteriors of one source from those of another source. The medians of these differences (one median for each site) were then averaged across sites for each comparison. Width of 95% CI's were also compared between the different data sources for each transition date using paired *t*-tests.

## 3 Results

### 3.1 Overall fits

The selected phenological model fit well to the GOES daily data (Fig. 3a and b, Supplementary Figs. S3 and S4). Credible interval widths in the rescaled phenology fits were noticeably narrower for PhenoCam models, than GOES, and similar between the two MODIS products (Fig. 3a and b, Supplementary Figs. S5 and S6). Several sites (*e.g.,* Green Ridge 2018; Fig. 3e and g) had spring green-up periods that were shorter than the 16-day temporal resolution of the MODIS products, but were shown in the GOES measurements (Fig. 3e). Based on the output from the paired *t*-tests, PhenoCam transition date uncertainties were statistically narrower than those of all satellite data for all transition dates (*p*-value < 0.015 for all comparisons; Table S2). All GOES transition dates were statistically more certain (based on CI width) than the corresponding MODIS EVI estimates (*p*-value < 0.03), but were only significantly more certain than MODIS NDVI for middle and end of spring (Fig. 4; Supplementary Table S2).

### 3.2 Spring transition dates

GOES was correlated with PhenoCam for the start of spring transition, where MODIS possessed an early bias (Fig. 5; Table 3). GOES vs. PhenoCam (*i.e.,* PhenoCam median transition dates were the independent variable and GOES median transition dates were the dependent variable) had the highest $R^2$ values (0.62 versus 0.00 and 0.00) and the lowest RMSE and average bias (5.18 ± 0.03 earlier than PhenoCam; Table 3). Both MODIS NDVI and EVI, on the other hand, were biased earlier 10.18 ± 0.1 and 11.66 ± 0.03 days on average, respectively. This bias was consistent amongst most sites (Fig. 6a). GOES was also most correlated with PhenoCam for the middle of spring estimate, with the highest $R^2$ value (0.72), lowest RMSE (7.06 days) and lowest average bias of 0.92 ± 0.03 days earlier than PhenoCam (Table 3). Most of the MODIS NDVI and EVI models were biased early (Fig. 6b). Both MODIS data products were slightly more correlated with PhenoCam for the end of spring than GOES. MODIS NDVI had the highest $R^2$ value of 0.80, but had a slightly earlier bias than MODIS EVI (1.75 ± 0.04 days vs 0.79 ± 0.05 days). MODIS EVI and GOES had the same $R^2$ values of 0.71, but GOES had a later bias of 3.35 ± 0.03 days (Table 3). There existed less correlation between GOES and the MODIS products (Supplementary Table S3).

### 3.3 Autumn transition dates

Autumn transition dates agreed less across all sources of data than in Spring. Except for MODIS EVI at the end of autumn ($R^2$ of 0.36), none of the satellite-based data sources explained any variation in PhenoCam transition date estimates (Table 3). Both NDVI sources (*i.e.,* GOES and MODIS) were consistently biased later (Table 3). MODIS EVI was biased earlier than PhenoCam for the beginning and middle of autumn transition dates. Except for the middle of autumn where MODIS NDVI and GOES had a slight correlation ($R^2$ of 0.34), satellite data sources were overall uncorrelated with each other for all autumn transition dates (all had $R^2$ values = 0.00). The biases were the smallest between MODIS NDVI and GOES for the first two

transition dates (GOES was 5.25 ± 0.24 days earlier and 2.89 ± 0.05 days later than MODIS NDVI in the start and middle of Autumn, respectively; Supplementary Table S3).

## 4 Discussion

### 4.1 Spring transition dates similarity

Our hypothesis that GOES spring transition dates are more similar than MODIS to PhenoCam ones was supported by our results for the start and middle of spring. While MODIS earlier sensing start of spring compared to PhenoCam has also been attributed to a variety of reasons ranging from different viewing angles that sense the heterogeneity of phenological change between different canopy layers (Ahl et al., 2006; Andrew D. Richardson and O'Keefe, 2009; Keenan et al., 2014b; Ryu et al., 2014; Schwartz et al., 2002), spatial scaling affected by significant topography (Fisher et al., 2007), and snow melt (Delbart et al., 2006), all these issues should affect the GOES-PhenoCam comparison as well. Based off of the similarity found here between GOES and PhenoCam at marking the start and middle of spring, the mismatch between MODIS and PhenoCam is likely largely due to the temporal resolution of the MODIS products. Hufkens et al. (2012) also points out that the temporal resolution of MODIS products cannot be expected to precisely track rapid leaf emergence in the spring due to the longer temporal resolution. By averaging over a relatively long time period (compared to the length of spring green-up), such techniques likely prematurely inflate NDVI and EVI values, giving the false impression that green-up occurred earlier. GOES, on the other hand, allows for daily NDVI estimates that are inherently more capable of tracking the initial spring increase.

Contrary to our hypothesis, though, we found that both MODIS indices were slightly less biased than GOES with PhenoCam at the end of spring, even with a higher or similar $R^2$ value to MODIS NDVI and EVI, respectively. Bias is likely more reliable as a measure of similarity than $R^2$ and RMSE because it includes the uncertainties in the transition dates for each site by utilizing the MCMC posterior samples instead of just comparing the median transition dates for each site. The later bias of GOES compared to PhenoCam and the MODIS products could potentially be due to an early bias in both PhenoCam and the MODIS products. PhenoCam GCC has been found previously to reach its end of spring before many physiological traits such as total chlorophyll concentrations, leaf area and mass, leaf nitrogen and carbon concentrations, and leaf area index (Keenan et al., 2014b; Yang et al., 2014). Thus, inherent differences between GCC and NDVI could cause the slightly later bias of GOES to PhenoCam at the end of spring. This should not affect GOES's ability to monitor interannual variability in this transition date as more years become available. MODIS could be biased early for the end of spring in a similar way as that discussed in the previous paragraph related to its temporal resolution. Klosterman et al. (2014) found, however, that on average MODIS had a later end of spring estimate than PhenoCam. One primary component of their study was to investigate the impacts of land-cover heterogeneity on transition date comparisons between different sources and, thus, they included more heterogeneous landscapes in their dataset. They concluded that due to scaling issues, MODIS pixels that have smaller proportions of deciduous forests have MODIS end of spring estimates that are later than near-surface estimates. Thus, it is reasonable to assume that our attempt to not include any sites with substantial non-forest land cover types within the MODIS

and GOES pixels would lower our average bias. The effects of land-cover heterogeneity on the estimates of end of spring transition should be kept in mind when using GOES to monitor more heterogeneous sites.

As with many studies, the results and conclusions of this study could depend on the methods used. It is possible that a different transition date estimation method (*e.g.,* the one proposed by Klosterman et al., 2014, that also accounted for summer green-down, which we were not focused on) would result in different conclusions. If there is bias in the methods here to estimate transition dates, it is shared across data sources, sites, and years. Similarly, using a different MODIS product might also result in different conclusions. While a daily MODIS product does exist, it is also created using multi-day periods of measurements (Ju et al., 2010) and, thus, is also possibly subject to similar constraints. We encourage others to consider more complex models and other phenology products, but the primary aim of this study is to demonstrate the value of GOES for studying phenology with an initial comparison to PhenoCams and MODIS.

### 4.2 Spring and autumn compared

As we hypothesized, spring transition dates were more similar across data sources than autumn ones. This mismatch between PhenoCam GCC autumn transition dates and NDVI and EVI (low $R^2$ values and high biases) has been found in numerous other studies (*e.g.,* Hufkens et al., 2012; Keenan et al., 2014a; Klosterman et al., 2014; Richardson et al., 2018b; Zhang et al., 2018). This is most likely due to physiological differences between the different metrics (*i.e.,* GCC, NDVI, and EVI) that become more apparent in the autumn with changes in color and canopy structure often occurring separately. While all three metrics measure some combination of greenness and canopy structure, only GCC directly considers green reflectance in its calculation; leaf presence and canopy structure (Kobayashi et al., 2007; Pettorelli et al., 2005) have been found to impact NDVI and EVI more. The higher uncertainties in the autumn transition dates, compared to spring ones, across all data sources were expected given the longer season length and the higher heterogeneity in autumn compared to spring. For example, triggers of autumn phenology are less understood and consistent than spring (Piao et al., 2019) and the timing of autumn phenological events differs greater between species than in spring (Richardson et al., 2006). Like the other data sources, GOES spring transition date estimates were most certain and most similar to those derived from other data sources.

### 4.3 Autumn transition dates similarity

We hypothesized that in the autumn, the transition dates derived from GOES NDVI data are most similar to those from MODIS NDVI data, which was mostly true. The low biases that existed between the two at the start and middle of autumn (MODIS NDVI was $5.25 \pm 0.24$ days later and $2.89 \pm 0.05$ earlier than GOES, respectively; Supplementary Table S3) are promising while the high end of autumn bias (MODIS NDVI was $11.03 \pm 0.24$ days earlier than GOES) could potentially be due to the high amount of noise in the GOES data that remains in the winter in many sites (Supplementary Figs. S2 and S3). Future directions for GOES that should help decrease these biases include developing a snow cover mask (which is a planned GOES product), developing a more sophisticated atmospheric correction algorithm for GOES reflectance data, and developing methodology for correcting for seasonal variations in solar angle. GOES will inherently be better at establishing a winter

baseline at sites with less snowy days than sites that consistently have a layer of snow obstructing accurate satellite measurements. Developing multi-year phenology models to increase the number of winter observations by assuming the winter NDVI baseline is similar between years would improve this. Furthermore, more informative priors in the diurnal fit model for estimating daily NDVI values that change seasonally would also improve the ability to estimate winter NDVI values with more certainty. These will likely help improve the correlation between GOES NDVI and MODIS NDVI.

## 4.4 Uncertainty in transition date estimates

We hypothesized that the increased temporal frequency in GOES data would produce more certain estimates of transition dates than MODIS. In practice, we found differences between MODIS indices, with GOES transition dates being significantly more certain than MODIS EVI for all transition dates, but only significantly more certain than MODIS NDVI for the start and middle of spring. However, as previously discussed, there are nontrivial differences between *what* NDVI and GCC are measuring in the autumn and end of spring that transcend simple issues of data quality and quantity, which suggests GOES is providing important new information about vegetation phenology. Once future work further improves the GOES products, reducing noise due to factors such as snow and atmospheric attenuation, the widths of the CI's are expected to improve. Additionally, the lack of a spatially- and temporally-varying MODIS uncertainty product, as we have produced for GOES, provided a limitation to this comparison and it is possible that specific daily MODIS uncertainties, congruent to that we used from the GOES data, would affect this conclusion. In particular, many of the MODIS validation efforts have focused on within-season comparisons (*e.g.*, Miura et al., 2000) not periods of phenological transition and, thus, MODIS uncertainties are likely underestimated. Furthermore, the differences in results between comparing GOES transition dates' CI widths to those from MODIS NDVI and EVI is likely partially due to the differences in observational error applied. Based on Miura et al. (2000), a smaller observational error was applied to MODIS NDVI than MODIS EVI, which likely was enough to make all GOES transition date estimates significantly more certain than the respective MODIS EVI estimates, but not always MODIS NDVI. This emphasizes the importance of providing uncertainty estimates with remotely sensed phenology data, which fitting diurnal curves to GOES data provides (Wheeler and Dietze, 2019).

## 4.5 Future phenological applications of GOES NDVI data

With its full coverage and high temporal resolution, GOES-16 and GOES-17 have the potential to revolutionize the study of leaf phenology and allow for a variety of studies that previously would not have been possible at the extent they are now. First, many studies have found that climate change is altering phenology on the scale of days per decade (Cleland et al., 2007; Keenan et al., 2014a; Parmesan and Yohe, 2003; Root et al., 2003). The long temporal scale of the studied MODIS NDVI and EVI products limits their ability to precisely and accurately monitor both these trends and interannual variability. While a daily MODIS NDVI product is becoming more readily available (and MODIS measurements are taken sub-daily, but at varying viewing angles), it still remains more inaccessible than many of the lower-frequency MODIS data products because it is

relatively new. It is important to have additional remotely-sensed data sources, especially ones that are not affected by changing viewing angles.

Second, GOES provides real-time data of spring green-up even for those springs that occur quicker than the 16-day resolution of this MODIS product (*e.g.,* Green Ridge 2018 in Fig. 3). These sites possessed no 16-day MODIS NDVI nor EVI measurements during the green-up period. This limitation would become even more severe when monitoring of green-up in

real-time, as the transition would only be detected after the fact. The seasonal curve fitting methodology used here to estimate transition dates is not suitable in real-time, but alternative methods, such as determining that spring has started if the phenological index is 10% greater than the winter baseline or iteratively assimilating data into a process model (Viskari et al., 2015) could be used to provide insight into whether or not certain transition dates have occurred in real-time. With the data that GOES supplies, it becomes more possible to monitor and forecast the start and progression of green-up at large scales

using near-real-time data, instead of having to wait for the next reliable MODIS product value, which might be 15 days away.

A third beneficial future application of the high temporal GOES NDVI data is the ability to monitor the effects of storms (*e.g.,* hurricanes), droughts, and frosts on phenology and NDVI. It is possible that the effects of some of these disturbances are only present for less than the temporal resolution of MODIS. For example, Richardson et al. (2018b) found that the effect of a spring frost event was visible within PhenoCam data, but not clearly visible within the 16-day MODIS data.

By providing higher temporal spring NDVI data, it is more likely that the effects of similar frost events could be observed at more areas that do not have a PhenoCam present.

A fourth benefit is that by combining data from multiple high-temporal sources (*i.e.,* PhenoCam and GOES), we now may ask questions related to differentiating the physiological impacts of phenological change in different indices and at different spatial scales. For example, with high-temporal NDVI data, we now can start asking questions about what specific

phenological processes control the rate of spring increase (*i.e.,* budburst vs. leaf expansion) and how these are affected by spatial scale. Similarly, combining PhenoCam and GOES data has the potential to help us better disentangle different autumn phenological processes (*i.e.,* leaf color change vs. leaf fall).

We are not suggesting that GOES should replace other phenology data sources, but that a combination of different data sources, which each have their own strengths and weaknesses, is beneficial. MODIS has a much longer record of data

than GOES and still remains an important source of information. Additionally, it has a different spatial scale that is between PhenoCam and GOES and a combination of the three could help answer spatial questions related to NDVI (*e.g.,* how does NDVI scale between canopy level to landscape level and how does this change seasonally?).

In conclusion, we have shown that GOES-16 and -17 possess great potential at enhancing the monitoring of leaf phenology, which will allow us to ask and answer new questions and improve our knowledge of this complicated, but important

aspect of ecology and environmental science.

**Code availability**

All code is available on Github at https://github.com/k-wheeler/NEFI_pheno/tree/master/GOES_PhenologyPaper_Code (see the main file GOES_Phenology_Paper_Code.Rmd) and

https://github.com/k-wheeler/NEFI_pheno/tree/master/PhenologyBayesModeling.

**Data availability**

All data was publicly available and can be accessed as described in the methods section.

**Author contribution**

KW and MD designed the study and KW executed the modeling and analysis. KW wrote the manuscript with inputs and suggestions from MD throughout the writing process.

**Competing interests**

The authors declare that they have no conflict of interest.

**Acknowledgements**

This work was made possible by the U.S. National Science Foundation grant 1638577. KIW also acknowledges support under the U.S. National Science Foundation Graduate Research Fellowship grant number 1247312. Special thanks to the Dietze lab
members for feedback on the manuscript. Any opinion, findings, and conclusions or recommendations expressed in this material are those of the authors and do not necessarily reflect the views of the National Science Foundation. We thank our many collaborators, including site PIs and technicians, for their efforts in support of PhenoCam. The development of PhenoCam has been funded by the Northeastern States Research Cooperative, NSF's Macrosystems Biology program (awards EF-1065029 and EF-1702697), and DOE's Regional and Global Climate Modeling program (award DE-SC0016011). We
acknowledge additional support from the US National Park Service Inventory and Monitoring Program and the USA National Phenology Network (grant number G10AP00129 from the United States Geological Survey), and from the USA National Phenology Network and North Central Climate Science Center (cooperative agreement number G16AC00224 from the United States Geological Survey). We also thank the USDA Forest Service Air Resource Management program and the National Park Service Air Resources program for contributing their camera imagery to the PhenoCam archive. Research at the Bartlett
Experimental Forest tower is supported by the National Science Foundation (grant DEB-1114804) and the USDA Forest Service's Northern Research Station. Research at the Coweeta flux tower is funded through the USDA Forest Service, Southern

Research Station; USDA Agriculture and Food Research Initiative Foundational Program, award number 2012-67019-19484; EPA agreement number 13-IA-11330140-044; and the National Science Foundation, Long-Term Ecological Research (LTER) program, award #DEB-0823293. Research at Harvard Forest is partially supported through the National Science Foundation's

LTER program (DEB-1237491), and Dept. of Energy Office of Science (BER). The Hubbard Brook Ecosystem Study is a collaborative effort at the Hubbard Brook Experimental Forest, which is operated and maintained by the USDA Forest Service, Northern Research Station, Newtown Square, PA. Research at the MOFLUX site is supported by the U.S. Department of Energy, Office of Science, Office of Biological and Environmental Research Program, Climate and Environmental Sciences Division. ORNL is managed by UT-Battelle, LLC, for the U.S. Department of Energy under contract DE-AC05-00OR22725.

U.S. Department of Energy support for the University of Missouri (Grant DE-FG02-03ER63683) is gratefully acknowledged. Research at the Morgan-Monroe Ameriflux site is supported by the US Departement of Energy, Office of Science, Office of Biological and Environmental Research throuth the Ameriflux Management Project administered by Lawrence Berkeley National Lab. Funding for the Shenandoah PhenoCam and related research has been provided by the U.S. Geological Survey Land Change Science Program (Shenandoah National Park Phenology Project) with logistical support from the National Park

Service in collaboration with the University of Virginia Department of Environmental Sciences. Camera images from the Shining Rock Wilderness are provided courtesy of the USDA Forest Service Air Resources Management Program. Primary support for the University of Michigan AmeriFlux Core Site (US-UMd) provided by the Department of Energy Office of Science. Infrastructure support provided by the University of Michigan Biological Station. Research at the Willow Creek Ameriflux core site is provided by the Dept. Of Energy Office of Science to the ChEAS Cluster

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

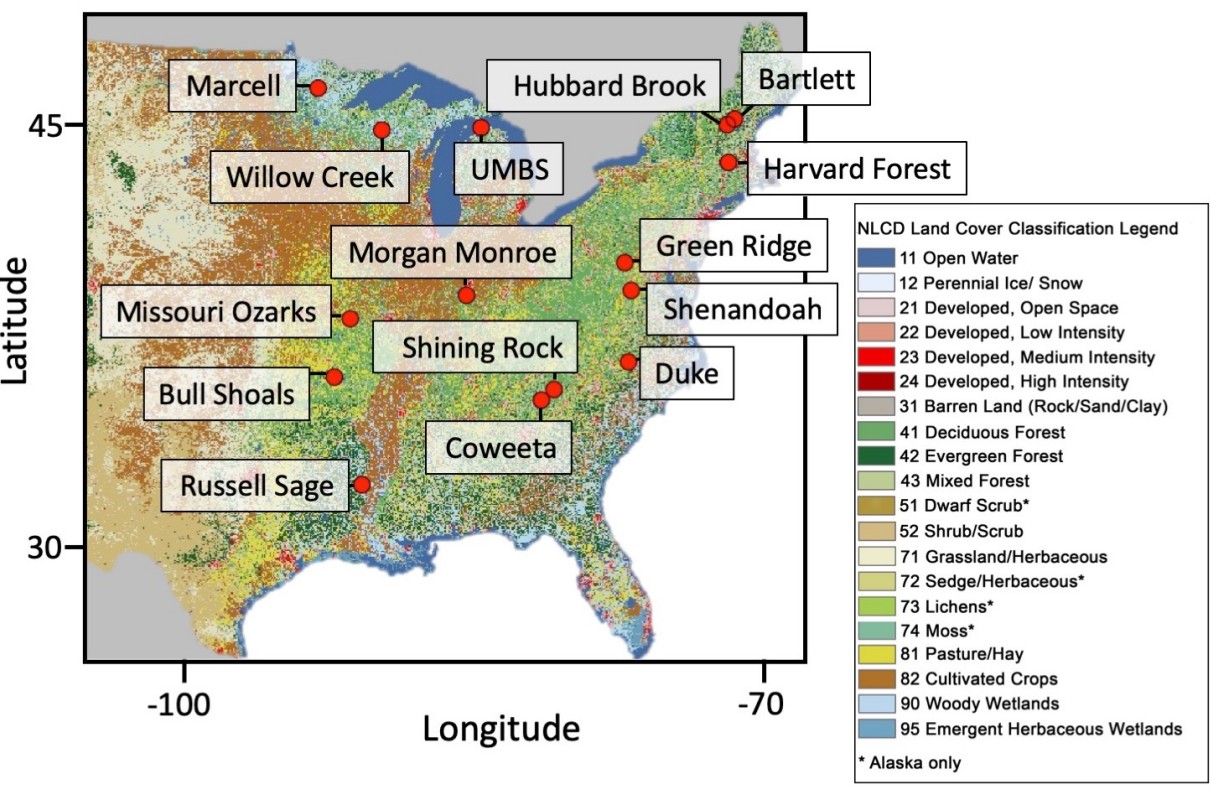

**Figure 1: Map of 2016 National Land Cover Database (NLCD) classification (Jin et al., 2019; Yang et al., 2018) for the study region showing the locations of the selected sites. Selected sites are located throughout the deciduous forested area in the United States. Specific locations are given in Table 1.**

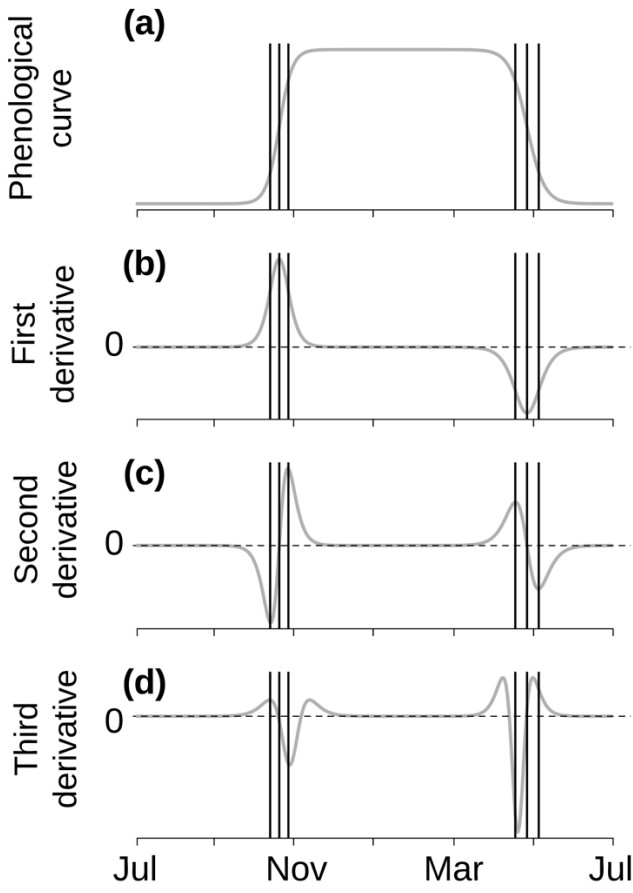


**Figure 2: Schematic based off of Fig. 2 in Zhang et al. (2003) describing the selection of transition dates (shown using the vertical lines). (a) illustrates an example phenological curve of green-down and green-up. (b) illustrates the first derivative. (c) illustrates the second derivative where the root (second derivative equals 0) gives the value of the middle of season date. (d) illustrates the third derivative where the roots give the start and end of both seasons.**

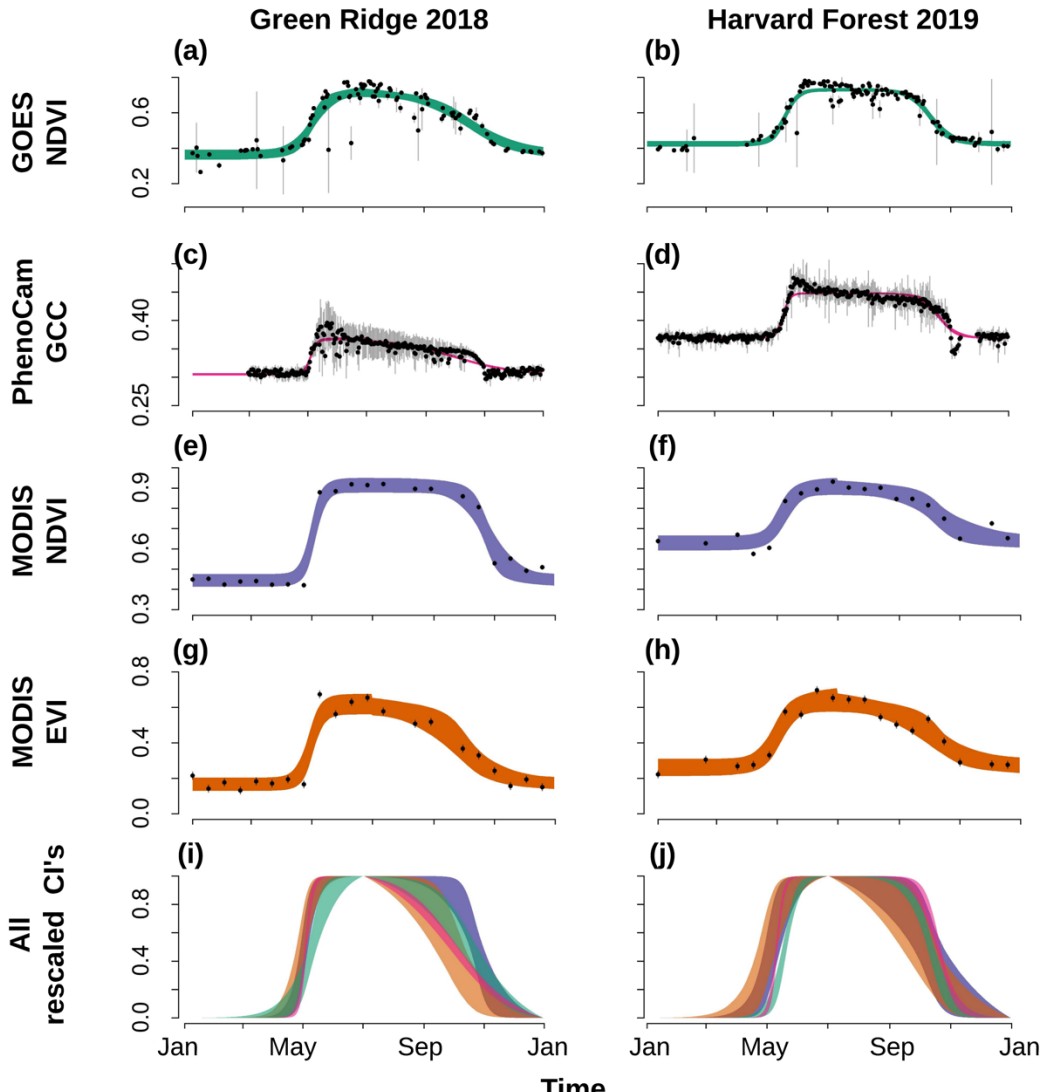


**Figure 3: Timeseries of data from GOES NDVI daily data (a and b), PhenoCam GCC (c and d), MODIS NDVI (e and f), MODIS EVI (g and h) and the rescaled credible intervals (i and j) for Green Ridge 2018 (left column) and Harvard Forest 2019 (right column). (a–h) include mean observations in black dots with 95% confidence intervals shown with vertical gray lines. 95% credible intervals (CI) are given with the different shading specific to each data source. The Green Ridge 2018 spring occurred quicker than**
**the MODIS temporal resolution, but was captured by the daily resolution of the GOES data.**

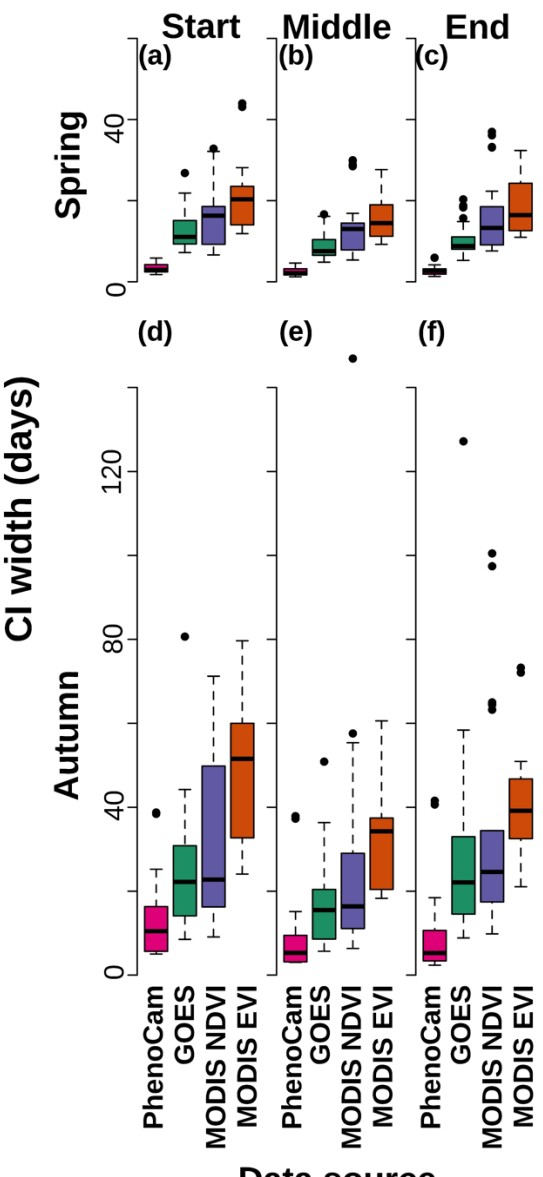

**Figure 4: The 95% credible interval (CI) widths for the different data sources for spring start (a), middle (b), and end (c); and autumn start (d), middle (e), and end (f). Colors denote the different data sources, which are labeled on the *x*-axis. PhenoCam had the most certain transition date estimates and GOES was always more certain than MODIS EVI, but only more certain than MODIS NDVI for the middle and end of spring.**


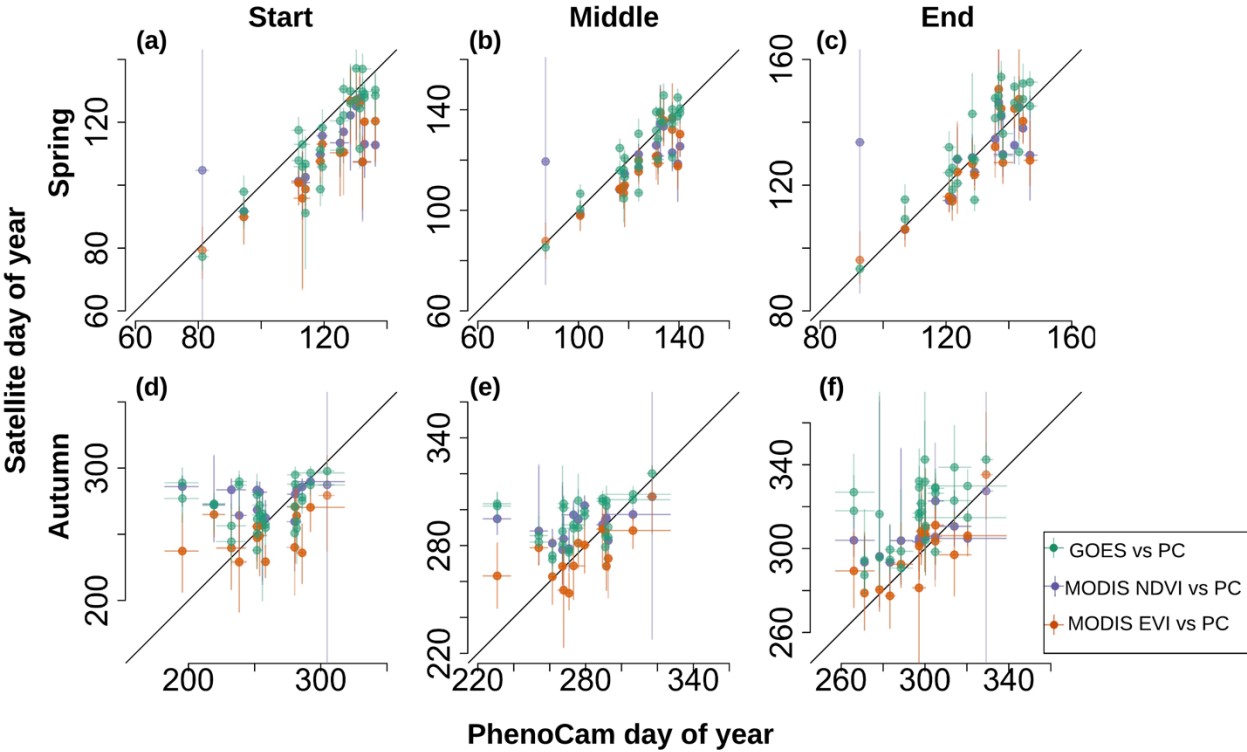

**Figure 5:** Scatter plots showing how the different data sources compare for their estimation of spring start (a), middle (b), and end (c); and autumn start (d), middle (e), and end (f). Median transition dates are indicated by the point and 95% credible intervals are indicated by the lines. The *x*-axis gives the day of year of the PhenoCam (PC) transition and the *y*-axis indicates the day of year of the different satellite data sources, which are color-coded as indicated in the legend. Spring correlations are much higher than autumn ones and GOES dates are more correlated at the start and middle of spring (a and b), but are slightly biased late at the end of spring (c).


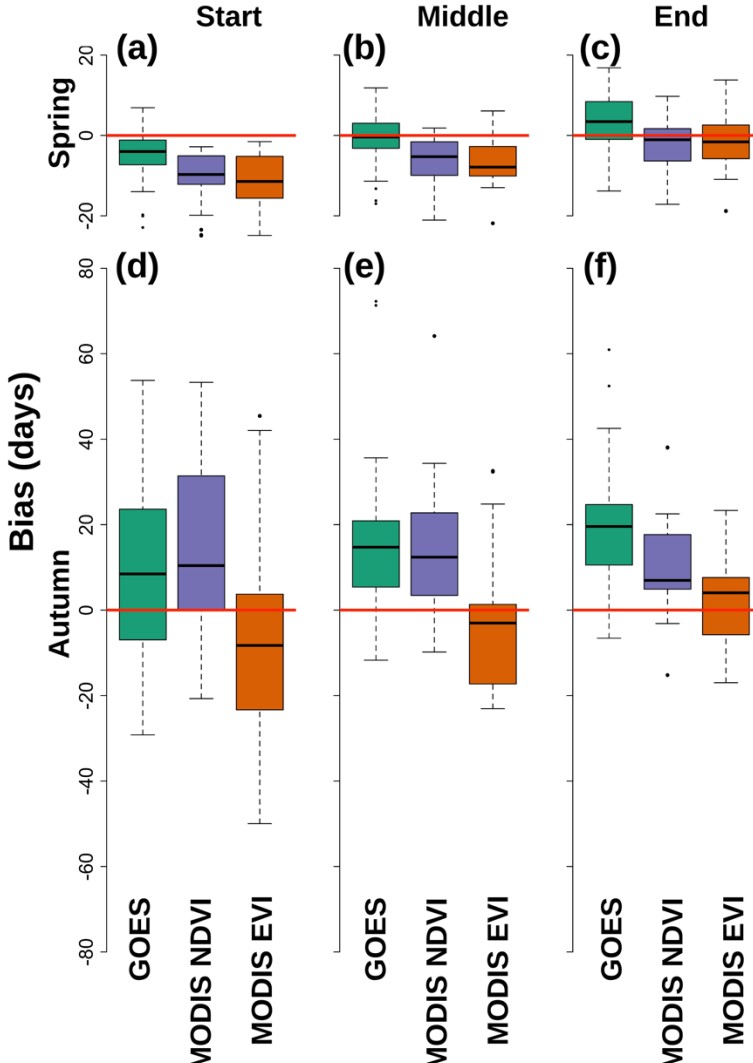

**Figure 6: The different biases for the different satellite-based data sources compared to PhenoCam for their estimation of spring start (a), middle (b), and end (c); and autumn start (d), middle (e), and end (f). A negative bias indicates the given data source was earlier than PhenoCam. The red line denotes zero bias. Boxes are color coded by the data source as indicated on the *x*-axis. The biases are larger in the autumn than in the spring. There are some with little median bias.**

**Table 1: Characteristics of selected sites including the coordinates and some climate data from WorldClim (Hijmans et al., 2005), which was assessed from the PhenoCam website (https://phenocam.sr.unh.edu/webcam/).**

| Site name | Latitude | Longitude | Mean Annual temperature (°C) | Mean Annual precipitation (mm) |
| --- | --- | --- | --- | --- |

| Marcell | 47.514 | -93.469 | 2.9 | 687.0 |
|---|---|---|---|---|
| Willow Creek | 45.806 | -90.079 | 3.9 | 820.0 |
| University of Michigan Biological Station (UMBS) | 45.560 | -84.714 | 5.9 | 797.0 |
| Bartlett | 44.065 | -71.288 | 5.5 | 1224.0 |
| Hubbard Brook | 43.927 | -71.741 | 4.6 | 1190.0 |
| Harvard Forest | 42.538 | -72.172 | 6.8 | 1139.0 |
| Green Ridge | 39.691 | -78.407 | 10.5 | 935.0 |
| Morgan Monroe | 39.323 | -86.413 | 11.2 | 1087.0 |
| Missouri Ozarks | 38.744 | -92.200 | 12.4 | 974.0 |
| Shenandoah | 38.617 | -78.350 | 8.4 | 1222.0 |
| Bull Shoals | 36.563 | -93.067 | 13.9 | 1084.0 |
| Duke | 35.974 | -79.100 | 14.6 | 1166.0 |
| Shining Rock | 35.390 | -82.775 | 9.3 | 1835.0 |
| Coweeta | 35.060 | -83.428 | 12.5 | 1722.0 |
| Russell Sage* | 32.457 | -91.974 | 18.1 | 1341.0 |

*Note: Due to the cessation of PhenoCam data collection in 2019, Russell Sage was only included in the 2018 analysis.

**Table 2: Parameter priors. "Satellites" refers to all GOES, MODIS NDVI and MODIS EVI. The priors on when the middle of spring and autumn occurred was set as the day of year (DOY). Normal distribution is abbreviated with N.**

| Parameter Name | Parameter Abbreviation | Data Source | Distribution (Mean, Standard Deviation) |
|---|---|---|---|
| Middle of spring (DOY) | $M_S$ | All | N (110, 40) |
| Middle of autumn (DOY) | $M_A$ | All | N (300, 40) |
| Spring rate of change | $b_S$ | All | N (-0.10, 0.05) |
| Autumn rate of change | $b_A$ | All | N (0.10, 0.05) |
| Minimum of phenological curve | $d$ | Satellites | N (0.6, 0.2) |
| Minimum of phenological curve | $d$ | PhenoCam | N (0.35, 0.15) |
| Range of phenological data | $c$ | Satellites | N (0.4, 0.2) |
| Range of phenological data | $c$ | PhenoCam | N (0.3, 0.15) |


**Table 3. Summary statistics for comparisons with PhenoCam transition dates. *Negative indicates the data source is earlier than PhenoCam. The widths of the 95% credible intervals (CI) of the biases are given.**

| Data Source | $R^2$ | RMSE (days) | Average Bias* (days; 95% CI) |
|---|---|---|---|
| *Start of Spring:* | | | |
| GOES | 0.62 | 9.06 | -5.18 ± 0.03 |
| MODIS NDVI | 0.00 | 13.53 | -10.18 ± 0.1 |
| MODIS EVI | 0.00 | 13.4 | -11.66 ± 0.03 |
| *Middle of Spring:* | | | |
| GOES | 0.77 | 7.06 | -0.92 ± 0.03 |
| MODIS NDVI | 0.00 | 10.86 | -5.56 ± 0.04 |
| MODIS EVI | 0.5 | 9.42 | -6.61 ± 0.03 |
| *End of Spring:* | | | |
| GOES | 0.71 | 7.99 | 3.35 ± 0.03 |
| MODIS NDVI | 0.12 | 10.49 | -0.95 ± 0.1 |
| MODIS EVI | 0.71 | 7.7 | -1.57 ± 0.03 |
| | | | |
| *Start of Autumn:* | | | |
| GOES | 0.00 | 32.3 | 12.59 ± 0.11 |
| MODIS NDVI | 0.00 | 34.2 | 17.84 ± 0.26 |
| MODIS EVI | 0.00 | 26.67 | -7.5 ± 0.11 |
| *Middle of Autumn:* | | | |
| GOES | 0.00 | 25.61 | 17.04 ± 0.07 |
| MODIS NDVI | 0.00 | 23.43 | 14.16 ± 0.08 |
| MODIS EVI | 0.00 | 15.64 | -2.7 ± 0.07 |
| *End of Autumn:* | | | |
| GOES | 0.00 | 26.58 | 21.5 ± 0.07 |
| MODIS NDVI | 0.00 | 16.17 | 10.47 ± 0.24 |
| MODIS EVI | 0.36 | 11.23 | 2.11 ± 0.06 |

