# Peer review of "Improving the monitoring of deciduous broadleaf phenology using the Geostationary Operational Environmental Satellite (GOES) 16 and 17"

_Biogeosciences, 2020_

## Referee Comment (RC1) · Anonymous Referee #1 · 4 Sep 2020

General comments: This article presents a novel application of Wheeler's algorithm for estimating midday NDVI from GOES data to estimate the dates of phenological transitions. Using phenocam data as a validation dataset, the authors test the correspondence of key phenological transitions as derived from GOES vs MODIS NDVI or EVI data, with the hypotheses relating to the higher frequency of GOES vs MODIS data, as well as sensitivities of the different indices to leaf presence vs color. Encouragingly - estimates of two spring transition dates do seem to align better for the GOES data than MODIS. and in some cases the GOES algorithm produces estimates with smaller

[Figure]

CIs. However, this was not the case for all phenophases and it is difficult to discern why. There are many differences between the datasets used including the algorithms used for preprocessing as well as the temporal frequencies and bands which make interpretation difficult. Overall, I found the paper interesting but perhaps better suited to a more remote-sensing oriented journal as it is difficult to distill a strong biological story in the comparisons presented.

Specific comments: Equation 3: I think that the top case should be for t>=k and the bottom for t<k

Did you investigate the 0.6040 artefact in the raw bands rather than just the indices? I wonder if it is also creating less obvious errors in other parts of the data - e.g. in S2 there are some outliers just after the series of values that were removed. Are both the red and NIR bands fixed at some value for these and are those invalid NIR/red values whereever the occur or only when they contribute to a ratio of 0.6040?

Would there be a way to automate the removal of the 68 days to further automate the algorithm?

Better legends on the supplemental material would improve readability.

The authors assert the GOES data can provide real-time estimates of phenological transitions whereas MODIS cannot due to the temporal frequency. However as I understand it the algorithm was fit on the entire time series at once, not subsetting down particular portions of the year. It is possible that the higher temporal frequency of the GOES data would allow for real-time estimation, but I think to demonstrate this one would need to iteratively refit using data only up to Jan, Feb, March (for example) and compare the estimated transition dates between the two datasets/algorithms. It is possible none would fit, though adding an informed prior on c and d based on the year(s) prior data might(?) provide enough information for them to converge without the full season of data.

[Figure]

What would implications be of NOT using the 16 composited NDVI/EVI from modis and instead using the native temporal resolution data and preprocessing as per the GOES? Would using the same snowcover mask on both datasets align the results? Is there any way to make the raw data more comparable to disentangle the cause of the differences (bands vs temporal resolution vs pre-processing)?

---

## Referee Comment (RC2) · Anonymous Referee #2 · 7 Sep 2020

Wheeler and Dietze presented a new satellite dataset to monitor vegetation phenology of deciduous broadleaf forests in the United States. The authors compared ground-based PhenoCam observations with new phenology observations from GOES 16 & 17 and MODIS vegetation indices (MOD13Q1). The authors found that the high-temporal-resolution (sub-daily) GOES product produces estimations of the phenological transition dates with higher confidence (narrower confidence interval) compared with MODIS NDVI and EVI data (16-day resolution). The GOES product correlates with Phenocam observation well in the spring, but less so in the fall.

Overall, this is a useful contribution to the community. It represents a new way to monitor vegetation phenology. The results are robust, and the manuscript is mostly well-written. I have some reservations regarding the discussion on the advantages of GOES vs. MODIS, which I will describe below.

The authors stated that the benefit of the GOES series is the high temporal resolution (sub-daily). The authors used the comparison to the PhenoCam data (and better performance of the GOES data) to support this claim. This claim to me is mostly correct with one exception: the comparison between GOES time-series and MODIS 16-day time-series is not fair because there is a daily MODIS reflectance product at 500 m resolution that takes into account of the impact of sun-sensor geometry – the MOD43A4 product. A quick search using google scholar showed a few papers that use daily MODIS product for phenological studies:

Liu, Y., Hill, M.J., Zhang, X., Wang, Z., Richardson, A.D., Hufkens, K., Filippa, G., Baldocchi, D.D., Ma, S., Verfaillie, J. and Schaaf, C.B., 2017. Using data from Landsat, MODIS, VIIRS and PhenoCams to monitor the phenology of California oak/grass savanna and open grassland across spatial scales. Agricultural and Forest Meteorology, 237, pp.311-325.

Ju, J., Roy, D.P., Shuai, Y. and Schaaf, C., 2010. Development of an approach for generation of temporally complete daily nadir MODIS reflectance time series. Remote Sensing of Environment, 114(1), pp.1-20.

Keenan, T.F., Gray, J., Friedl, M.A., Toomey, M., Bohrer, G., Hollinger, D.Y., Munger, J.W., O'Keefe, J., Schmid, H.P., Wing, I.S. and Yang, B., 2014. Net carbon uptake has increased through warming-induced changes in temperate forest phenology. Nature Climate Change, 4(7), pp.598-604.

The results have demonstrated that GOES ABI is likely better than MODIS 16-day products in capturing spring phenology. Still, proper discussion (and some texts in the introduction) on the daily MODIS product is necessary. The downside of the MODIS

16-day product could potentially be addressed with a daily MODIS product.

A few detailed comments:

Line 51: Adding a sentence on why changing viewing angle limits the temporal resolution could be useful for non-remote-sensing readers.

Line 65: "subject to the same temporal limitations" – but both are subject to cloud impact. No data when there is cloud cover.

Line 144: This approach misses two main features in the GCC curve: The peak in the later spring and early summer, and the gradual decline in GCC in the summer to fall. Similarly, the equation does fit the NDVI data well (misses the summer decline in NDVI). See the following paper and the Klosterman et al. that was cited:

Elmore, A.J., Guinn, S.M., Minsley, B.J. and Richardson, A.D., 2012. Landscape controls on the timing of spring, autumn, and growing season length in mid‐A tlantic forests. Global Change Biology, 18(2), pp.656-674.

Using the methods from Elmore et al. or Klosterman et al. could potentially improve the late spring transition date estimation (and the mismatch between satellite the PhenoCam data).

Line 230: the use of "prematurely" gives a sense of that MODIS incorrectly estimates the start of spring. However, another possibility is the green-up of the understory may cause MODIS vegetation indices to increase even when Phenocam data do not show any changes. Richardson and O'Keefe (2009) showed that understory spring is about 10-20 days earlier at Harvard Forest:

Richardson, A.D., and O'Keefe, J., 2009. Phenological differences between understory and overstory. In Phenology of ecosystem processes (pp. 87-117). Springer, New York, NY.

Line 264: It is unclear what the difference is between "canopy greenness" and "leaf

presence and canopy structure". Both GCC and vegetation indices (EVI and NDVI) are affected by these factors, but in different ways: GCC (R, G, B) can be affected by these factors differently compared with EVI and NDVI (which has a NIR band). I suggest rewriting this sentence.

Section 4.2.: the uncertainties in the fall phenology estimation could also be attributed to the heterogeneity in the timing of fall phenology (compared with much-synchronized spring phenology). Worth some discussion.
* * *

---

## Author Comment (AC1) · 23 Oct 2020

Kathryn I. Wheeler and Michael C. Dietze

kiwheel@bu.edu

We want to thank the reviewer for their useful and thoughtful comments, as well as their overall positive response in feeling that our paper was interesting and a useful contribution to the community.

We appreciate that the reviewer thinks that this article may be better suited in a remote-sensing journal, but think that the biogeosciences community would be interested in a new way to monitor phenological change. We have already published the methodological details of our GOES algorithm in a remote sensing journal, and to make this paper more accessible to non-remote-sensors we will reduce the remote-sensing jargon. Ultimately our interest here is in the biological process of phenology and not the satellite specifics.

We thank the reviewer for catching the error in Equation 3. We will change that.

We thank the reviewer for the suggestion on how to further eliminate noise by looking at the raw red and near-infrared (NIR) values for the observations that had an NDVI value of 0.6040. We will update the manuscript to further explain that all of the NDVI values of 0.6040 only occurred in the early morning and the evening. Based on the days and sites that we looked at, the NDVI value of 0.6040 seems to occur for multiple different combinations of red and NIR values (e.g., R=1.87e-5, NIR=7.57e-5; R=-2.80e-4, NIR=-1.14e-3; R=3.18e-4, NIR=1.29e-3; R=1.95e-5, NIR=7.90e-5). It seems that when these reflectances occur they are usually in pairs that result in an NDVI value of 0.6040. When we saw them occurring elsewhere it was limited to the noisy morning and evening sections. We already filtered most of this out by removing the NDVI values that occurred before 1.5 hours after sunrise or 1.5 hours before sunset. In future work, we will consider further filtering based on the reflectance bands.

We agree that as this research scales up it will become more important to come up with better quality control methods. There are definitely options for how to do that, but implementing and testing these QC algorithms, and selecting between different approaches, is beyond the scope of this paper, especially since we wanted to maintain the focus on phenology and not on the remote sensing methods.

We thank the reviewer for pointing out the ambiguity in the supplementary figures. We think that labeling the axes would help make them clearer and plan to do that.

We thank the reviewer for pointing out that our assertion that the higher temporal frequency of GOES data would provide better real-time estimates of transition dates is not fully supported by our methods. We will change the wording for this in the manuscript

to focus more on GOES providing real-time estimates of phenological conditions. We'll also expand the Discussion on how that real-time data does or does not translate into better estimates of transition dates and how that might be affected by different estimation methods (e.g., concluding spring has started when the index value is 10% greater than the winter baseline). We are currently working on iterative fitting and the estimation of phenological transition dates in near-real-time and will include this in future papers.

The raw data are not comparable between GOES and MODIS because GOES is geostationary and MODIS is on sun-synchronous satellites. Unlike GOES, MODIS changes viewing angles between measurements. The advantage of geostationary data is not just the higher frequency of measurements, but also the consistent viewing angle. The MODIS composite product is created not just to average to reduce noise, but to consider the multiple viewing angles of MODIS observations. To the best of our knowledge there are also no good error estimates for the daily MODIS data, which would also affect our ability to compare them. There has been a lot of calibration and validation that has gone into the MODIS products so returning to the raw MODIS data would be difficult and very time consuming – at the moment we don't think this would be worth the effort.

---

## Author Comment (AC2) · 23 Oct 2020

We want to thank the reviewer for their useful and thoughtful comments, as well as their overall positive response in feeling that our paper was interesting and a useful contribution to the community.

While a daily MODIS product is available and some studies do use it, it is harder to use and less straight-forward to access. The MODIS product used in this paper still represents the most commonly used product in most phenological studies. We will add

more to the discussion pointing out that there are other MODIS products that might result in different conclusions.

Line 51: We agree that more explanation of the limitations that changing viewing angle has would be beneficial to the readers of this journal.

Line 65: We will add something about clouds limiting both; however, with the higher temporal frequency clouds limit GOES less because there are more opportunities for clear observations. Indeed, in our previous paper (Wheeler and Dietze 2019 Remote Sens.) we showed that GOES was frequently able to produce robust estimates of daily NDVI even when mid-day observations (the most common timing for polar orbits) were obscured.

Line 144: We were aware of the Klosterman et al. (2014) paper and their findings. We did not choose to use their equation because it has substantially more parameters in it that were not explained in the manuscript and we could not determine reasonable priors for most of them. We did already mention their estimation of transition dates in the methods section of our paper, but will add some more to the discussion. We acknowledge that with the transition date estimation methods that we are using we are missing the gradual decline in the summer, but we are not trying to estimate that. The bias in the transition date estimations is shared across all of the curves and sources in our study. Overall, would encourage others in the community to consider more complex models, but feel that fitting a range of alternative models goes beyond our primary aim of providing a demonstration of the value of GOES for studying phenology and an initial comparison to Phenocams and MODIS.

Line 230: We will remove the word "prematurely" and specifically reference understory vs top-of-canopy instead of only hinting to it.

Line 264: We thank the reviewer for pointing out the ambiguity in this sentence and we will revise it.

Section 4.2: We agree that heterogeneity in the timing of fall phenology is important and will add some more discussion about it.

---

## Author Response (AR1)

Reviewer 1:

- Overall, I found the paper interesting but perhaps better suited to a more remote-sensing oriented journal as it is difficult to distill a strong biological story in the comparisons presented.
    - We appreciate that the reviewer thinks that this article may be better suited in a remote-sensing journal, but think that the biogeosciences community would be interested in a new way to monitor phenological change. We have already published the methodological details of our GOES algorithm in a remote sensing journal, and to make this paper more accessible to non-remote-sensors we will reduce the remote-sensing jargon. Ultimately our interest here is in the biological process of phenology and not the satellite specifics.
- Equation 3: I think that the top case should be for t>=k and the bottom for t<k
    - Changed. (Line 174).
- Did you investigate the 0.6040 artefact in the raw bands rather than just the indices? I wonder if it is also creating less obvious errors in other parts of the data - e.g. in S2 there are some outliers just after the series of values that were removed. Are both the red and NIR bands fixed at some value for these and are those invalid NIR/red values whereever the occur or only when they contribute to a ratio of 0.6040?
    - We thank the reviewer for the suggestion on how to further eliminate noise by looking at the raw red and near-infrared (NIR) values for the observations that had an NDVI value of 0.6040. We updated the manuscript (line 135) to further clarify that all of the NDVI values of 0.6040 only occurred in the early morning and the evening. Based on the days and sites that we looked at, the NDVI value of 0.6040 seems to occur for multiple different combinations of red and NIR values (*e.g.*, R=1.87e-5, NIR=7.57e-5; R=-2.80e-4, NIR=-1.14e-3; R=3.18e-4, NIR=1.29e-3; R=1.95e-5, NIR=7.90e-5). It seems that when these reflectances occur they are usually in pairs that result in an NDVI value of 0.6040. When we saw them occurring elsewhere they were still limited to the noisy morning and evening sections. We already filtered most of this out by removing the NDVI values that occurred before 1.5 hours after sunrise or 1.5 hours before sunset. In future work, we will consider further filtering based on the reflectance bands, but almost all of this noise appeared to be already filtered out by removing the morning and evening values.
- Would there be a way to automate the removal of the 68 days to further automate the algorithm?
    - Added sentence on lines 153-155.
- Better legends on the supplemental material would improve readability.
    - We added axes to the supplementary figures and updated legends.
- The authors assert the GOES data can provide real-time estimates of phenological transitions whereas MODIS cannot due to the temporal frequency. However as I understand it the algorithm was fit on the entire time series at once, not subsetting down particular portions of the year. It is possible that the higher temporal frequency of the GOES data would allow for real-time estimation, but I think to demonstrate this one would need to iteratively refit using data only up to Jan, Feb, March (for example) and compare the estimated transition dates between the two datasets/algorithms. It is possible none would fit, though adding an informed prior on c and d based on the year(s) prior data might(?) provide enough information for them to converge without the full season of data.

- o We thank the reviewer for pointing this out – it is definitely true that this specific curve-fitting approach would not be the best option for real-time applications. We clarified this by slightly changing abstract (line 21) and adding a sentence to the discussion (lines 357-360) that points out alternative approaches, such as detecting the passage of predefined thresholds or more sophisticated iterative data assimilation and forecasting approaches, like we have previously employed in Viskari et al. 2015 Ecological Applications. Indeed, adding GOES to our iterative phenology forecast was the original motivation for much of this work.

- What would implications be of NOT using the 16 composited NDVI/EVI from modis and instead using the native temporal resolution data and preprocessing as per the GOES? Would using the same snowcover mask on both datasets align the results? Is there any way to make the raw data more comparable to disentangle the cause of the differences (bands vs temporal resolution vs pre-processing)?
  - o The raw data are not comparable between GOES and MODIS because GOES is geostationary and MODIS is on sun-synchronous satellites. Unlike GOES, MODIS changes viewing angles between measurements. The advantage of geostationary data is not just the higher frequency of measurements, but also the consistent viewing angle. The MODIS composite product is created not just to average to reduce noise, but to consider the multiple viewing angles of MODIS observations. To the best of our knowledge there are also no good error estimates for the daily MODIS data, which would also affect our ability to compare them. There has been a lot of calibration and validation that has gone into the MODIS products so returning to the raw MODIS data would be difficult and very time consuming – at the moment we don't think this would be worth the effort.

Reviewer 2:

- The authors stated that the benefit of the GOES series is the high temporal resolu- tion (sub-daily). The authors used the comparison to the PhenoCam data (and better performance of the GOES data) to support this claim. This claim to me is mostly correct with one exception: the comparison between GOES time-series and MODIS 16-day time-series is not fair because there is a daily MODIS reflectance product at 500 m resolution that takes into account of the impact of sun-sensor geometry – the MOD43A4 product. The results have demonstrated that GOES ABI is likely better than MODIS 16-day products in capturing spring phenology. Still, proper discussion (and some texts in the introduction) on the daily MODIS product is necessary. The downside of the MODIS 16-day product could potentially be addressed with a daily MODIS product.
  - o We updated the abstract (lines 14-15) to emphasize that these results are for the studied MODIS products. We added more to the introduction (lines 60-65) on why we're focusing on the 16-day product and to the discussion (lines 291-295).
- Line 51: Adding a sentence on why changing viewing angle limits the temporal resolu- tion could be useful for non-remote-sensing readers.
  - o Added at lines 54-59.

- Line 65: "subject to the same temporal limitations" – but both are subject to cloud impact. No data when there is cloud cover.
  - We added a sentence pointing out that because GOES makes measurements throughout the day it is better poised to be able to capture times within a day that may be cloud free (lines 87-88). In our previous work, which is mentioned here, we showed that we could estimate daily NDVI reliably on most days even if there was cloud cover at a specific time (e.g. noon) by exploiting the predictable diurnal pattern of GOES NDVI. Obviously, there are still days with complete cloud cover, but these are much fewer for GOES than MODIS.
- Line 144: This approach misses two main features in the GCC curve: The peak in the later spring and early summer, and the gradual decline in GCC in the summer to fall. Similarly, the equation does fit the NDVI data well (misses the summer decline in NDVI). See the following paper and the Klosterman et al. that was cited:
  - Elmore, A.J., Guinn, S.M., Minsley, B.J. and Richardson, A.D., 2012. Landscape controls on the timing of spring, autumn, and growing season length in mid-Atlantic forests. Global Change Biology, 18(2), pp.656-674.
  - Using the methods from Elmore et al. or Klosterman et al. could potentially improve the late spring transition date estimation (and the mismatch between satellite the Phe- noCam data).
  - We added sentences to discussion (lines 288-291) pointing to this as a future direction, but also noting that any biases are likely shared across the different sensors.
- Line 230: the use of "prematurely" gives a sense of that MODIS incorrectly estimates the start of spring. However, another possibility is the green-up of the understory may cause MODIS vegetation indices to increase even when Phenocam data do not show any changes. Richardson and O'Keefe (2009) showed that understory spring is about 10-20 days earlier at Harvard Forest:
  - We changed "prematurely" to "earlier" (line 257) and further clarified this in lines 258-259.
- Line 264: It is unclear what the difference is between "canopy greenness" and "leaf presence and canopy structure". Both GCC and vegetation indices (EVI and NDVI) are affected by these factors, but in different ways: GCC (R, G, B) can be affected by these factors differently compared with EVI and NDVI (which has a NIR band). I suggest rewriting this sentence.
  - We tried to clarify this in lines 301-303.
- Section 4.2.: the uncertainties in the fall phenology estimation could also be attributed to the heterogeneity in the timing of fall phenology (compared with much-synchronized spring phenology). Worth some discussion.
  - Added sentences to lines 305-307.

[revised manuscript text omitted]

---

## Author Response (AR2)

Dear Dr. Paul Stoy:

Thank you, and Referee 3, for your continued feedback on our manuscript. We have revised the manuscript based on the Referee 3's suggestions. We added a paragraph about the limitations of our NDVI values to lines 163—168. We also tried to simplify the language throughout including the specific passages mentioned by the reviewer (*e.g.,* lines 6; 8—10; 20; 84; 87; 289—291; 297-298; 325—326; 349—350; 374—376). We also changed "noise" to "erroneous data" on line 158; and added "at the time of writing" to the PhenoCam site-year number on line 55. We did not change "continental" to either of the other two words suggested by the reviewer as this is the word used in the GOES R Series Product Definition and Users' Guide for the used CONUS product. We also did not change our text about the spatial averaging of the red pixels when calculating NDVI. Because we are aggregating the raw red data to the same scale as the raw NIR, which is a linear operation, before calculating NDVI (which is a nonlinear operation) there is no violation of Jensen's Inequality. Indeed, the approach suggested by Referee 3 (calculate NDVI at the red-pixel scale using the non-disaggregated NIR) is the violation of Jensen's because the later aggregation of these 4 NDVI values would involve averaging a nonlinear transform (the NDVI calculation). We appreciate your consideration of our manuscript for publication.

Sincerely,
Kathryn Wheeler